# Remdesivir and dexamethasone as tools to relieve hospital care systems stressed by COVID-19: A modelling study on bed resources and budget impact

Guillaume Béraud[1]*, Jean-François Timsit[2,3], Henri Leleu[4]

**1** Infectious Diseases Department, University Hospital of Poitiers, Poitiers, France, **2** APHP- Bichat Hospital—Medical and Infectious Diseases Intensive Care Unit, Paris, France, **3** IAME UMR 1137 Université de Paris (Paris-Diderot), Paris, France, **4** Public Health Expertise, Paris, France

\* beraudguillaume@gmail.com

## Abstract

Remdesivir and dexamethasone are the only drugs providing reductions in the lengths of hospital stays for COVID-19 patients. We assessed the impacts of remdesivir on hospital-bed resources and budgets affected by the COVID-19 outbreak. A stochastic agent-based model was combined with epidemiological data available on the COVID-19 outbreak in France and data from two randomized control trials. Strategies involving treating with remdesivir only patients with low-flow oxygen and patients with low-flow and high-flow oxygen were examined. Treating all eligible low-flow oxygen patients during the entirety of the second wave would have decreased hospital-bed occupancy in conventional wards by 4% [2%; 7%] and intensive care unit (ICU)-bed occupancy by 9% [6%; 13%]. Extending remdesivir use to high-flow-oxygen patients would have amplified reductions in ICU-bed occupancy by up to 14% [18%; 11%]. A minimum remdesivir uptake of 20% was required to observe decreases in bed occupancy. Dexamethasone had effects of similar amplitude. Depending on the treatment strategy, using remdesivir would, in most cases, generate savings (up to 722€) or at least be cost neutral (an extra cost of 34€). Treating eligible patients could significantly limit the saturation of hospital capacities, particularly in ICUs. The generated savings would exceed the costs of medications.

## Introduction

The most prominent burden of the COVID-19 outbreak in France is the saturation of hospital resources due to the massive influx of COVID-19 patients at hospital. Non-pharmaceutical interventions (NPIs) have been shown to be effective in containing the spread of the outbreak. Additionally, no pharmaceutical drugs with proven efficacy in containing the spread of SARS-CoV-2 were available at the early phase of the pandemic.

Therefore, most public health policies initially consisted of NPIs aimed at "flattening the curve" (ranging from physical distancing to school closures and full lockdowns) to avoid

**Funding:** HL is an employee of Public Health Expertise, which received funding from GILEAD for this study. The funders had no role in study design, data collection and analysis, decision to publish, or preparation of the manuscript.

**Competing interests:** I have read the journal's policy and the authors of this manuscript have the following competing interests: - GB participated to advisory boards and gave lectures in symposia for Gilead. - JFT reports participation to advisory board for Gilead related to covid-19 antiviral therapies. - HL is employed by PHE which received funding from Gilead. This does not alter our adherence to PLOS ONE policies on sharing data and materials.

hospital saturation and triage of incoming patients in the intensive care unit (ICU). However, the benefits of the strictest interventions (curfews and lockdowns) to limit hospital saturation have to be balanced with the subsequent economic [1], social and medical consequences, such as reduced access to care for cancer patients [2] and consultations for heart attack and myocardial infarctions [3].

At the beginning of November 2020, only two approved anti-COVID-19 therapies, dexamethasone and remdesivir, had demonstrated a significant reduction in the length of hospital stay among COVID-19 patients [4, 5].

We hypothesized that the use of these two therapies would result in reducing bed occupancy, hence playing a role in limiting the saturation of the healthcare system and diminishing the need for the strictest NPIs, such as lockdowns and curfews. However, the effect size of the use of these two therapies has been assessed only in patients participating in randomized controlled trial (RCT) studies. Hitherto, no real-life study has gauged the extent to which the use of these two drugs would help to reduce bed occupancy. In this regard, modelling studies represent a useful tool for addressing the lack of real-life studies while evaluating medical interventions.

To assess to what extent pharmaceutical interventions could play a role in limiting hospital saturation, we used a recently published agent-based model [6] to precisely estimate the impact of remdesivir and dexamethasone, alone or in combination, on COVID-19 patient length of stay in both conventional wards and ICUs, during the second wave of the COVID-19 pandemic in France.

## Methods

The primary objective of this epidemiological modelling study was to estimate the impact of a shorter time to recovery and a reduction in the risk of requiring high-flow or invasive ventilation associated with the use of remdesivir and dexamethasone in patients hospitalized for COVID-19 on (1) hospital-bed occupancy in a conventional ward and (2) ICU-bed occupancy. Our secondary objective was to determine the minimum uptake of remdesivir required to observe a significant impact on bed occupancy.

The model was combined with epidemiological data concerning the current second wave of the COVID-19 outbreak in France, as well as with data from RCTs for parameters for which real-life data were not available (specifically, data concerning treatment efficiency).

### Model structure

We used a previously published epidemiological model that was calibrated and validated for the French setting [6]. The model is a stochastic agent-based model that includes (i) a realistic synthetic population generated with demographic characteristics, comorbidities and household structure representative of the French population [7–10], (ii) social contacts among the individuals in the population including intrafamilial, school or work contacts, friends or extended family members (at home or at bars and restaurants), and those made while grocery shopping, riding public transport and performing cultural activities, and (iii) a SARS-CoV-2 disease model, which translates the social contacts into infection probability and simulates the patient's pathway from infection to recovery. Compared with the previous publication [6], the risk of contamination and the percentage of asymptomatic patients were updated to take into account the SARS-CoV-2 seroprevalence in May 2020 in France [11], which was previously unknown. Parameters are detailed in S1 Table in S1 Appendix.

## Hospital and ICU admission and length of stay

The disease model includes the probabilities of hospital admission, ICU admission and death. These probabilities were stratified by age and adjusted for comorbidities, including obesity, diabetes, chronic cardiac diseases, and chronic respiratory diseases, based on hazard ratios calculated using data from Institut Pasteur [12] and a large cohort study [13]. The risk of death was reduced by an average of 15% in July, and the ICU duration was reduced by 30% to fit to the observational data, thus reflecting the overall improvement in patient care observed in France, including but not limited to the use of dexamethasone [14]. Delays between infection, symptom onset, hospital admission, ICU admission, death or recovery were based on early studies [12, 15–17] and were updated according to a recent analysis [14]. Consequently, we assumed that hospitalized patients would follow one of several possible pathways [14] depending on disease severity and oxygen requirements, thus resulting in different lengths of stay. Most patients (62.6%) required only a short hospitalization (average of 8 days), with a small subset of patients (7.7%) requiring long-term rehabilitation care after hospitalization. Among the patients requiring short hospitalization, most presented with mild or moderate symptoms and needed at most low-flow oxygen. Based on the patient distribution of the ACTT-1, it was assumed that 21% would not require oxygen at any point during their stay. The remaining 29.6% of patients were considered to have severe symptoms. Of these, 47.6% died without being admitted to the ICU, thus resulting in 84.4% of the patients having never been admitted to the ICU. Of the remaining 15.6% of patients, 63% were admitted directly to the ICU for high-flow oxygen or mechanical ventilation, and 37% were initially admitted to a conventional hospital bed with low-flow oxygen before being secondarily admitted to the ICU for high-flow oxygen or mechanical ventilation after a few days following a worsening of their condition. Based on observed data [14], the average duration in the ICU was estimated to be 16 days, with a subset of individuals (16.6%) having an average stay of 35 days. For modelling purposes, we summarized and translated these literature-based pathways into a probability tree diagram (Fig 1).

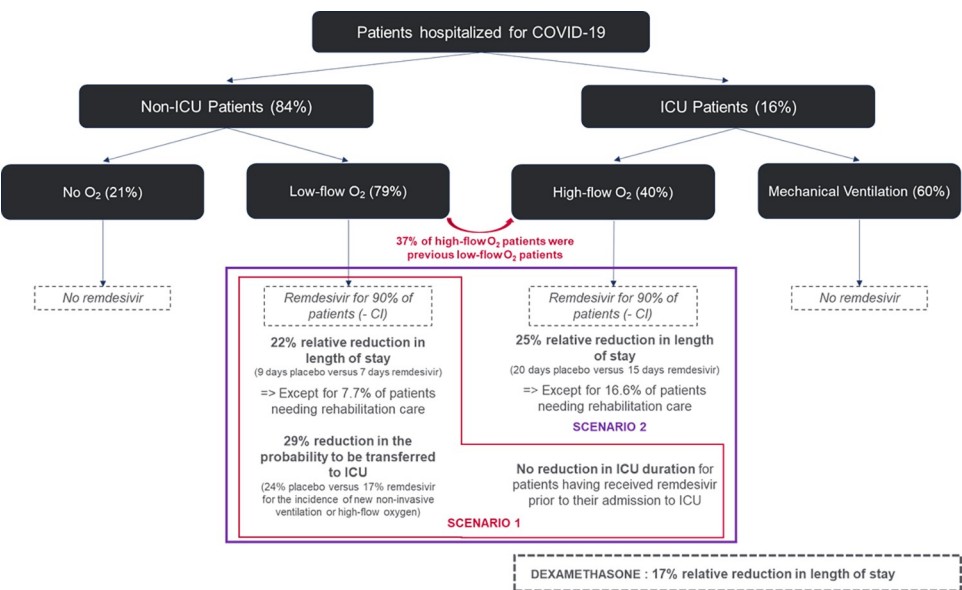

**Fig 1. Probability tree diagram representing the patient pathways.**

## Impact of remdesivir and dexamethasone

The impact of remdesivir on the lengths of hospital stay was extrapolated from the ACTT-1 [5]. Initially, a scenario was tested in which remdesivir would be given only to all eligible low-flow-oxygen patients, as recommended by the French Ministry of Health when providing the vial granted by the European commission [18] and the French Haute Autorité de Santé, (HAS) [19] [scenario a]. The model included both patients requiring low-flow oxygen during their stay and patients initially treated with low-flow oxygen before being admitted to the ICU following a worsening of their condition. In addition, we conservatively assumed that up to 10% of patients would not receive remdesivir because of renal or hepatic contraindications.

The effect size of remdesivir on the reduction in the length of stay was based on the results in the subgroup of hospitalized patients receiving oxygen who showed a reduction in the time to recovery from a median of 9 to 7 days, a relative reduction of 22%. This reduction was applied as a relative effect on the length of hospital stay simulated for each patient receiving remdesivir in the model to ensure transferability of the results to the French population. As remdesivir is currently used for less than 2% of French patients, we assumed that hospital duration in the model for current standard of care did not include a remdesivir effect. To be conservative, it was assumed that patients requiring long-term rehabilitation care after hospitalization in the observational data (approximately 7.7% of the low-flow-oxygen patients who were discharged alive) would not see a reduction in their lengths of stay. This figure matches the percentage of patients who were still alive and hospitalized at 28 days in the remdesivir arm of the ACTT-1 (7.7%), for which there are no data on the reduction in lengths of stay.

We considered that treating low-flow-oxygen patients with remdesivir reduces the risk of needing high-flow or invasive ventilation and only applied this effect to the patients who were initially admitted with low-flow oxygen before being transferred to the ICU following a worsening of their condition. For these patients, receiving remdesivir was associated with a 29% reduction in the probability of being admitted to the ICU based on a reduction from 24% to 17% in the incidence of new non-invasive ventilation or high-flow oxygen in the remdesivir group compared to the placebo group in the ACTT-1 [5]. We assumed that for those with a worsening of symptoms, every patient with low-flow oxygen would initially be treated with high-flow or non-invasive ventilation before, being switched to invasive ventilation if necessary. In addition, we assumed no reduction in ICU duration for patients who had received remdesivir prior to their ICU admission. Both of these assumptions resulted in conservative estimates of the remdesivir effect on the length of stay in the ICU.

Second, we tested a scenario where remdesivir was given to all eligible low-flow- and high-flow-oxygen patients as recommended by the official German guidelines [20] and according to the EMA market authorization label [scenario b].

It was assumed that high-flow-oxygen patients represented 40% of all patients in the ICU based on the ACTT-1 patient distribution [5]. Similar to that on low-flow-oxygen patients, the effect size of remdesivir on high-flow-oxygen patients' length of stay was based on ACTT-1 data. In the subgroup of patients receiving high-flow oxygen or non-invasive ventilation, the observed length of stay decreased from 20 days to 15 days, a relative reduction of 25%. Similar to low-flow-oxygen patients, no reduction in length of stay was applied for high-flow-oxygen patients needing rehabilitation care (16.6% of the high-flow-oxygen patients in the observational data).

According to the RECOVERY trial [4], we assumed that dexamethasone had an impact on length of stay only for patients admitted to the ICU, with a 17% reduction in the ICU length of stay. This was based on the differences in the proportions of patients discharged from the hospital within 28 days conditional on being alive between the dexamethasone and placebo

groups. We assumed that the increase in the proportion of patients discharged within 28 days translated directly to a reduction in the length of stay. No average length of stay was reported for the RECOVERY trial [4].

The main assumptions for treatment effects and patient distribution are presented in Fig 1.

## Cost of remdesivir and hospital stay

Remdesivir costs were considered for an average duration of treatment, taking into account discontinuation due to hospital discharge. Based on the results of the simulation, the average duration was 4 days (€1725, €345 x 5 doses), which is in accordance with what was observed in clinical trials [21]. We also included an analysis with 5 full days of treatment (€2070, €345 x 6 doses) as recommended on the label.

We used diagnosis-related group (DRG)-based national hospital tariffs to estimate the daily cost of inpatient care in conventional and ICU wards [22]. Because COVID-19 is an emergent disease, there were no data on costs of care during hospitalization. For reimbursement of COVID-19-related stays, hospitals have used a generic DRG covering respiratory inflammation and infections (GHM 04M07) with an average daily cost of €550.55 for conventional wards and €1690.65 for the ICU after adjusting for inflation. However, this DRG does not cover all the additional expenses related to COVID-19 care incurred by hospitals, as many of them ended up incurring deficits while exhibiting maximum capacities. For this reason, hospitals received 377 million euros in late April [23] to cover these additional expenses for stays between early March and mid-April. Based on the number of patients hospitalized in France during the same period [24], this amounts to an average additional cost of €227.70 per day for conventional wards and €699.24 for ICU wards, totalling an average daily cost of care for COVID-19 of €778.25 for a conventional stay and €2,389.89 for an ICU stay. However, these figures are unlikely to represent the full cost of COVID-19 hospital stays. Indeed, an extra payment of 2.4 billion euros was provided for hospitals to cover COVID-19 expenses in the 2021 Social Security financing bills. In September 2020, the initial amount provisioned was 1 billion euros to compensate for the excessive costs of the first wave and was then revised upward to anticipate the consequences of a second wave [25].

## Statistical analysis

The stochastic agent-based microsimulation model was run from 01/03/2020, until 01/02/2021, on 500,000 individuals. The results were based on an average of 200 simulations and were extrapolated to the French population, which comprises 67 million people. In addition, we used a bootstrap procedure, a technique based on the sampling with a replacement for a large number of iterations to provide standard error or bias estimates on parameters [26]. We provided uncertainty measures by using 200 bootstrap samples based on the random variations in all non-calibrated parameters simultaneously, either within the 95% confidence intervals for parameters estimated from the literature or within a +/-20% interval if the parameter was assumed. All results are presented per 100,000 inhabitants.

First, we ran a scenario reflecting the evolution of the COVID-19 epidemic in France between 01/03/2020 and 18/11/2020, and examined whether the model had adequate calibration, i.e., whether it was able to adequately reproduce hospital admissions, hospital-bed occupancy, ICU admissions, ICU-bed occupancy and daily mortality observed in France (S1 Fig in S1 Appendix).

The scenario was based on a full lockdown between 17/03/2020 and 11/05/2020, followed by a progressive return to pre-pandemic social contacts between 11/05/2020 and 01/07/2020 except for schools remaining closed and 30% telework. We assumed, based on current

epidemiological trends, a seasonal variation in SARS-CoV-2 transmission, with a 23% reduction between 01/07/2020 and 25/09/2020 [27]. Indeed, a notable drop in temperature occurred in France on the 25/09/2020, which was followed by a significant increase in disease transmission. On 01/09/2020, school resumed for all students, and telework was reduced to 16%. Face mask use at work, in public transport, while grocery shopping and for cultural events increased from 15% to 70% between 01/04/2020 and 01/09/2020 [28] and remained stable thereafter. We assumed that face masks were not used in households or with friends and extended family members. Curfew was instated on 17/10/2020, reducing contacts with friends and extended family members by half and cancelling all cultural events. Lockdown was instated again on 30/10/2020 except for school and work. We assumed that only 50% of individuals worked remotely. For this analysis, it is assumed that this second lockdown lasted 6 weeks and was followed by a curfew similar to the one instated on 17/10/2020 until February, with 30% of individuals remaining on telework.

Next, we examined the impact of the systematic introduction of remdesivir for low-flow-oxygen patients since 01/08/2020 on hospital-bed occupancy, ICU admissions and ICU-bed occupancy relative to the current treatment strategy. Additionally, we examined the impact of dexamethasone on ICU-bed occupancy. When assuming that dexamethasone is routinely used in current patient care, we estimated the impact by *removing* the impact of dexamethasone on current ICU-bed occupancy. Additionally, we estimated the impact of introducing remdesivir for different degrees of uptake (from 10% to 90%) and for whether or not high-flow-oxygen patients were also treated.

Finally, the budget impact compared the potential benefits of remdesivir use with its cost. We estimated the average cost of treatment per patient from the health insurance perspective with a time horizon limited to the inpatient stay, taking into account potential savings from shorter stays in both conventional and ICU wards.

For validation, curves of model-predicted and observed daily hospital and ICU admissions, as well as hospital and ICU-bed occupancy and daily mortality, were compared visually (S1 Fig in S1 Appendix). The model was implemented using C++. The source code is on a public repository on GitHub (https://github.com/henrileleu/covid19). In addition, for comparison with other countries, epidemiological curves of cases, tests and positivity rates from the start of the epidemic to 01/02/2021 are provided in S3 Fig in S1 Appendix.

## Results

The model shows that, compared with the current situation, using remdesivir would result in significantly lower hospital-bed occupancy in the conventional ward, lower ICU admissions and lower ICU-bed occupancy. Fig 2 presents the impact of using remdesivir between 01/08/2020 and 01/02/2021 on hospital-bed occupancy (A; B), ICU admissions (C; D) and ICU-bed occupancy when a) all eligible low-flow-oxygen patients are treated (E; F) and alternatively when b) all eligible low-flow- and high-flow-oxygen patients are treated (G; H).

Noticeably, on the day of the peak, using remdesivir would have resulted in a 6% lower hospital-bed occupancy in the conventional ward in both scenarios (33,500 vs 35,650 hospital beds occupied in France on the 18/11/2020, or 50.0 vs 53.2 per 100,000 inhabitants). Regarding ICU-bed occupancy on the day of the peak, scenario a, "all eligible low-flow-oxygen patients are treated", would result in a 12% lower value (5,695 vs 5,025 in France or 8.5 vs 7.5 per 100,000) and a 16% lower value in scenario b, "all eligible low-flow- and high-flow-oxygen patients are treated" (4,760 vs 5694 in France or 7.1 vs 8.5 per 100,000).

Based on the model, we estimated that treating all eligible patients with remdesivir in both scenarios could have reduced hospital-bed occupancy by 4% [2%; 7%] on average throughout

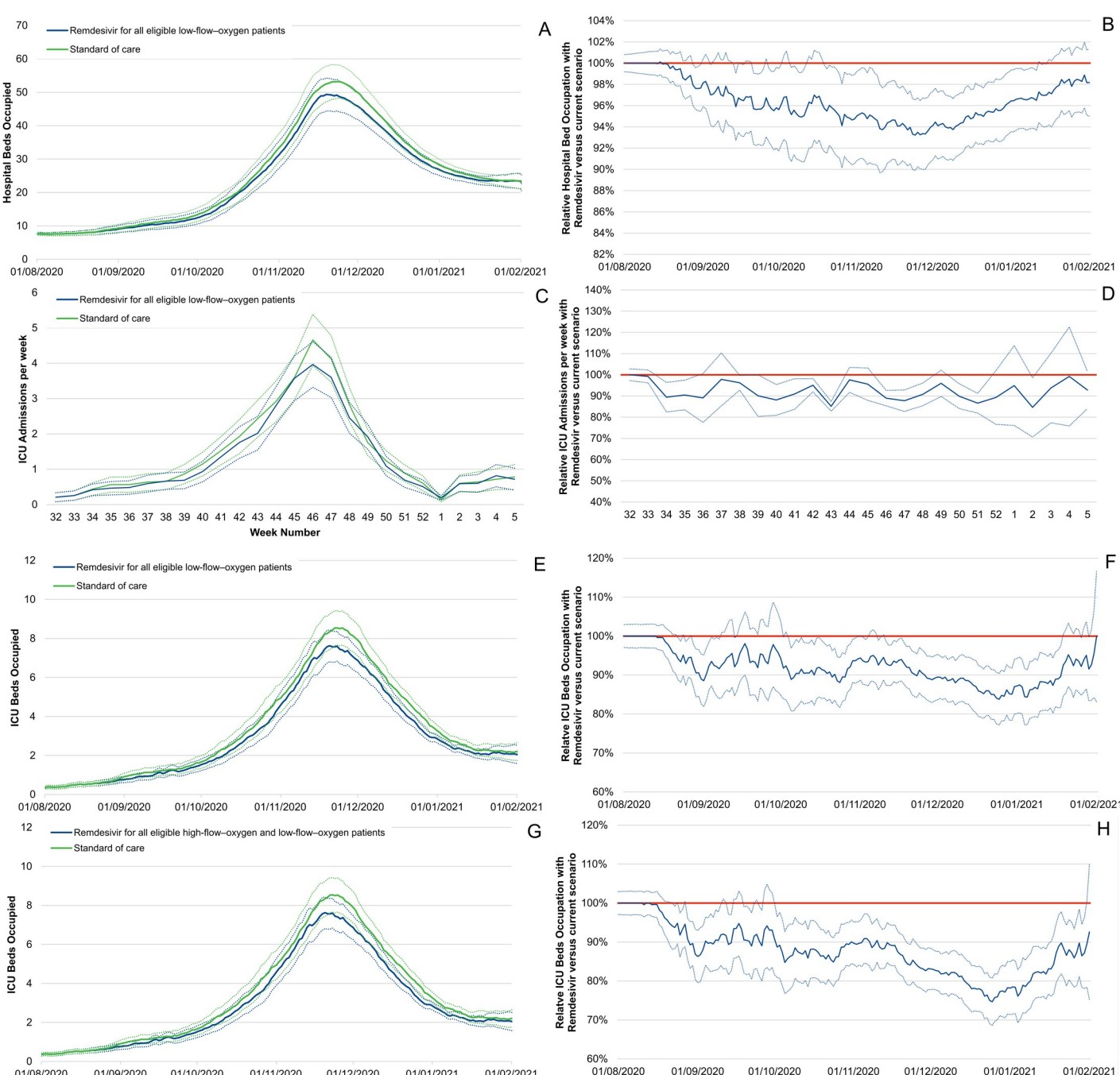

**Fig 2.** Estimations of the number of hospital beds occupied per day per 100,000 inhabitants (A), hospital bed occupancy associated with remdesivir use for all eligible low-flow-oxygen patients (scenario a) and all eligible low-flow and high-flow-oxygen patients (scenario b) relative to standard of care (B), ICU admissions per week per 100,000 inhabitants (C), ICU admissions associated with remdesivir use for all eligible low-flow-oxygen patients (scenario a) relative to standard of care (D), number of ICU beds occupied per day per 100,000 inhabitants (E) and ICU bed occupancy associated with remdesivir use for all eligible low-flow-oxygen patients (scenario a) relative to standard of care (F), number of ICU beds occupied per day per 100,000 inhabitants (G) and relative ICU bed occupancy associated with remdesivir use for all eligible low-flow- and high-flow-oxygen patients (scenario b) relative to standard of care (H) between 01/08/2020 and 01/02/2021. Remdesivir use has been limited to all eligible low-flow patients since 01/08/2020.

**Table 1. Comparison of the number of patients treated with remdesivir, number of hospital bed-days occupied, number of ICU admissions and number of ICU bed-days occupied between 01/08/2020 and 01/02/2021 if remdesivir had been used for all eligible patients with low-flow-oxygen therapy since 01/08/2020.** Ninety-five percent confidence intervals are estimated from 200 bootstrap simulations.

| | Current standard of care | Remdesivir for all eligible low-flow-oxygen patients | Difference vs current standard of care | Relative reduction vs current standard of care |
|---|---|---|---|---|
| **Number of patients treated with remdesivir** | | | | |
| 01/08/2020–01/02/2021 | No patients treated with remdesivir | 144.7 [134.5; 154.9] | 144.7 [134.5; 154.9] | |
| 30/10/2020–14/12/2021 | No patients treated with remdesivir | 67.97 [61.18; 74.77] | 67.97 [61.18; 74.77] | |
| **Average hospital beds occupied per day** | | | | |
| 01/08/2020–01/02/2021 | 25.03 [23.02;27.04] | 23.91 [21.96;25.87] | - 1.12 [- 1.67;- 0.57] | 0.96 [0.93;0.98] |
| 30/10/2020–14/12/2021 | 45.68 [40.82;50.54] | 43.02 [38.39;47.65] | - 2.66 [- 4.07;- 1.24] | 0.94 [0.91;0.97] |
| **Average ICU admissions per day** | | | | |
| 01/08/2020–01/02/2021 | 0.20 [0.19; 0.22] | 0.19 [0.17; 0.20] | - 0.02 [- 0.02;- 0.01] | 0.92 [0.89; 0.95] |
| 30/10/2020–14/12/2021 | 0.40 [0.36; 0.44] | 0.37 [0.33; 0.41] | - 0.03 [- 0.05;- 0.02] | 0.91 [0.88; 0.95] |
| **Average ICU beds occupied per day** | | | | |
| 01/08/2020–01/02/2021 | 3.32 [3.04; 3.61] | 3.02 [2.76; 3.28] | - 0.31 [- 0.42;- 0.19] | 0.91 [0.87; 0.94] |
| 30/10/2020–14/12/2021 | 6.99 [6.21; 7.77] | 6.38 [5.68; 7.09] | - 0.60 [- 0.93;- 0.28] | 0.91 [0.87; 0.96] |

Results are per 100,000 inhabitants. Remdesivir use has been limited to all eligible low-flow-oxygen patients since 01/08/2020.

the study period, and by 6% [9%; 3%] during the second peak between 30/10/2020 and 14/12/2020, thus corresponding to an average daily increase of 1.12 [0.57; 1.67] and 2.66 [1.24; 4.07] beds per 100,000 or 747.9 [378.9; 1,117.0] beds and 1,779 [830; 2,730] for France, respectively (Table 1).

Similarly, treating all eligible low-flow-oxygen patients with remdesivir (scenario a) would have reduced ICU-bed occupancy by 9% [6%; 13%] on average throughout the study period and by 9% [4%; 13%] during the second peak, thus corresponding to an average daily gain of 0.31 [0.19; 0.42] and 0.60 [0.28; 0.93] beds per 100,000 or 204.5 [125.0; 284.0] and 404.4 [185.3; 623.5] beds for France, respectively (Table 1). The lower ICU-bed occupancy in our model is related to a reduction in daily ICU admissions of 0.02 [0.01; 0.02] per 100,000 or 10.8 [6.9; 14.7] admissions for France between 01/08/2020 and 01/02/2021. During the second peak, a reduction in ICU daily admissions of 0.03 [0.02; 0.05] per 100,000 or 23.0 [12.4; 33.6] was estimated.

Comparing scenario a to b reveals that treating all eligible low-flow- and high-flow-oxygen patients with remdesivir significantly improves ICU-bed occupancy. Therefore, remdesivir use reduces ICU-bed occupancy by 14% [18%;11%], which corresponds to an average daily gain of 0.48 [0.36; 0.60] beds per 100,000, 321.3 [204.2; 402.4] beds for France, between 01/08/2020 and 01/02/2021. When considering the second lockdown (30/10/2020-14/12/2020), the average daily gain was 0.98 [0.65; 1.32] beds per 100,000, or 657.8 [434.4; 881.1] beds for France (Table 2).

By comparison, our model also suggests that dexamethasone has a similar effect size as remdesivir on ICU-bed occupancy (S2 Fig in S1 Appendix): the current use of dexamethasone reduced ICU-bed occupancy by 0.34 [0.25; 0.42] bed-days per 100,000 or 224.5 [167.4; 281.2] for France between 01/08/2020 and 01/02/2020, as well as 0.64 [0.41; 0.87] bed-days per 100,000 or 431.1 [276.9; 585.4] for France during the second peak (S2 Fig in S1 Appendix).

**Table 2. Estimations of the total number of ICU bed-days occupied between 01/08/2020 and 01/02/2021 associated with the use of remdesivir for all eligible patients with low-flow-oxygen therapy and for the use of remdesivir for all eligible patients with low-flow-and high-flow-oxygen therapy since 01/08/2020 relative to the current standard of care in France.**

|  | ICU beds occupied (Bed-days) | Absolute difference vs. current standard of care | Relative difference vs. current standard of care | Peak occupancy (number of beds /100,000 inhabitants) |
|---|---|---|---|---|
| **01/08/2020–01/02/2021** | | | | |
| Current standard of care | 3.32 [3.04; 3.61] | 0 | 0 | 8.2 [7.2; 9.1] |
| Remdesivir for all eligible low-flow-oxygen patients | 3.02 [2.76; 3.28] | - 0.31 [- 0.42;- 0.19] | 0.91 [0.87; 0.94] | 7.5 [6.6; 8.4] |
| Remdesivir for all eligible low-flow- & high-flow-oxygen patients | 2.85 [2.60; 3.09] | - 0.48 [- 0.60;- 0.36] | 0.86 [0.82;0.89] | 7.2 [6.3; 8.1] |
| **30/10/2020–14/12/2020** | | | | |
| Current standard of care | 6.99 [6.21; 7.77] | 0 | 0 | 8.2 [7.2; 9.1] |
| Remdesivir for all eligible low-flow-oxygen patients | 6.38 [5.68; 7.09] | - 0.60 [- 0.93;- 0.28] | 0.91 [0.87;0.96] | 7.5 [6.6; 8.4] |
| Remdesivir for all eligible low-flow- & high-flow-oxygen patients | 6.01 [5.34; 6.67] | - 0.98 [- 1.32;- 0.65] | 0.86 [0.82;0.90] | 7.2 [6.3; 8.1] |

Additionally, Fig 3 shows the size of hospital-bed and ICU-bed occupancy based on the proportion of low-flow-oxygen patients receiving remdesivir. The figure suggests that at least 20% of patients need to receive remdesivir to produce a sizable reduction in bed occupancy.

Finally, Table 3 shows the results of the budget impact analysis. Stopping treatment after discharge led to an average treatment duration of 4 days. We also presented results assuming that every patient would receive a full 5-day course. As a result, using remdesivir would generate, in every scenario except one, cost savings ranging from 307.30€ to 720.63€. In the remaining worst-case scenario, the use of remdesivir is cost neutral (+37.70€).

## Discussion

We showed that the use of remdesivir would have a significant impact on hospital- and ICU-bed occupancy, even when administered only to patients with low-flow-oxygen therapy. We purposefully modelled the impact of remdesivir for the entire second wave, based on current known epidemiological dynamics and limited assumptions. This provides insight into the expected impact of implementing this public health policy for an entire wave. We also provided a focus on the peak period, which corresponds to the 2nd lockdown. Moreover, we demonstrated that even a limited uptake of remdesivir of at least 20% would generate a sizable beneficial effect in reducing bed occupancy and the need for transfer in the ICU.

During the first wave of the pandemic, considerable effort led to a temporary increase in the number of beds available in the ICU from 5000 to more than 7000 (the maximum number of patients in the ICU for COVID-19 was 7019 on 08/04/2020)($\simeq$ 11 ICU beds for 100,000 inhabitants) [29]. The reduction in ICU beds resulting from the use of remdesivir would have prevented overwhelming the healthcare system as well as the highly demanding increase in ICU beds. With COVID-related ICU-bed occupancy at the peak decreasing to 7.5 per 100,000 inhabitants, remdesivir would have hindered the overstretching of ICU-bed availability.

Limiting the saturation of ICU capacities would be beneficial in two ways. First, it would prevent triage for worsening patients requiring the ICU [30]. Second, it would improve healthcare quality not only for COVID-19 patients but also for the many non-COVID-19 patients who have to postpone their hospital care. It should be noted that this latter effect goes far beyond the few weeks of the outbreak peak. A recent study indeed showed that cancer

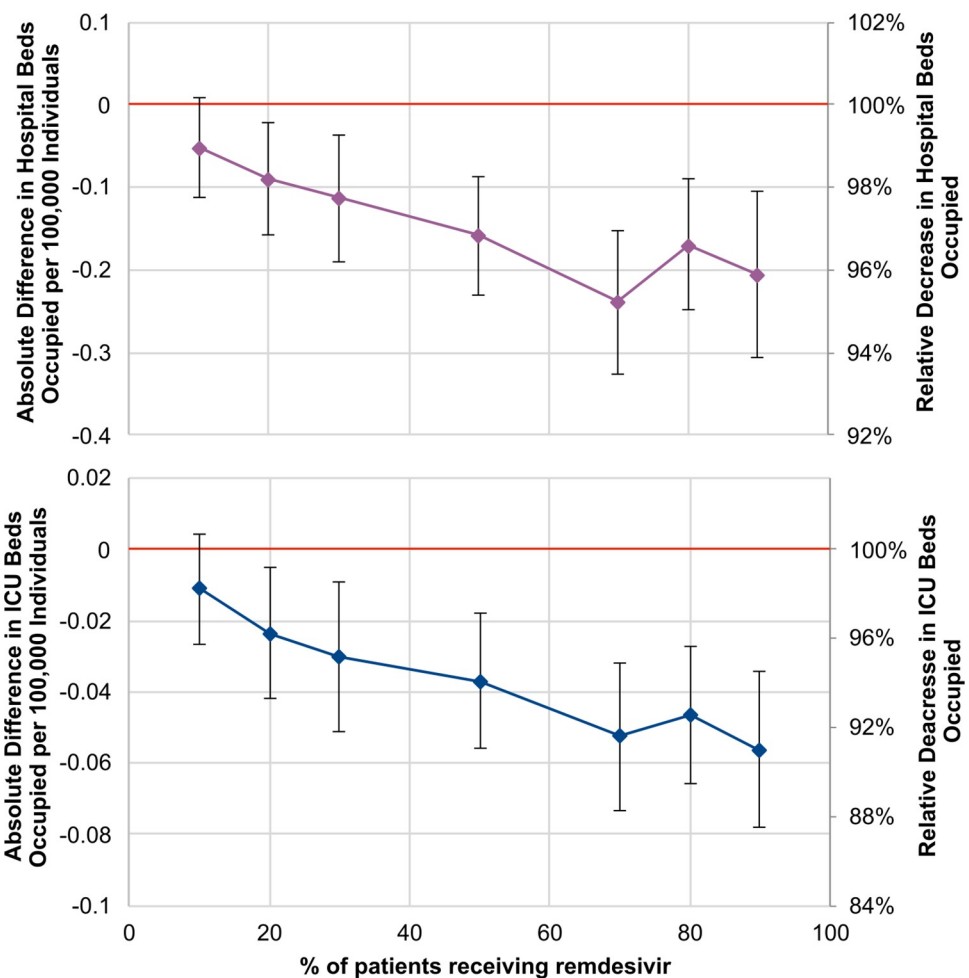

**Fig 3.** Estimations of the total number of hospital bed-days occupied (top) and total number of ICU bed-days occupied (bottom) between 01/08/2020 and 01/02/2021 associated with different rates of remdesivir use for all eligible patients with low-flow-oxygen therapy since 01/08/2020 relative to the current standard of care in France. Results are per 100,000 inhabitants. Remdesivir use has been limited to all eligible low-flow-oxygen patients since 01/08/2020. The stochastic nature of the model may be responsible for the small variations.

**Table 3. Impact of remdesivir use on budget.**

| | Remdesivir for all eligible low-flow-oxygen patients | | Remdesivir for all eligible low-flow- and high-flow-oxygen patients | |
|---|---|---|---|---|
| | Cost per patient (€) | | Cost per patient (€) | |
| Treatment duration | 4 days* | 5 days | 4 days* | 5 days |
| Average treatment cost per patient treated | 1725.00 | 2070.00 | 1725.00 | 2070.00 |
| Average avoided bed-days in standard care units per patient treated ** | -1104.66 | -1104.66 | -1054.52 | -1054.52 |
| Average avoided bed-days in ICU per patient treated | -927.64 | -927.64 | -1391.11 | -1391.11 |
| Savings generated with remdesivir use per patient treated | -307.30 | 37.70 | -720.63 | -375.63 |

\* Based on the results of the simulation, the average duration was 4 days (1725€, i.e., 345€ x 5 doses), which is in accordance with what was observed in clinical trials [21].

\*\* In both scenarios the absolute number of avoided occupied standard care beds is equal; however, more patients are treated in the "Remdesivir for all eligible low-flow- and high-flow-oxygen patients" group leading to a lower number of avoided occupied standard care beds per patient treated.

screenings have been postponed for many patients, which would necessarily result in diagnoses at more advanced stages with deleterious consequences [2].

We also showed that dexamethasone had an effect of similar amplitude, which justifies the wide use of both drugs.

As of November 2020, remdesivir use in France has been limited to patients requiring low-flow oxygen (i.e., oxygen delivery via a face mask from 3–15 l/min), but it has also been approved for patients receiving oxygenation delivered via high-flow nasal oxygenation systems. To put in perspective the result of implementing different public health guidelines, we assessed the benefit of extending remdesivir use to most of these severe patients who are in the ICU. Unsurprisingly, as this group of patients, representing 40% of ICU patients, benefited from a 25% shorter time to recovery, this benefit extension resulted in a noticeable effect on ICU-bed occupancy. Of note, this proportion of 40% was similar to the 35.2% that was seen in a prospective cohort study that included 4,643 patients from 138 hospitals in France, Belgium and Switzerland between February and May 2020 [31]. It is important to stress that the decrease in bed occupancy came from a shorter time to recovery but not from a higher rate of recovery, which was not significantly different between remdesivir and placebo for patients receiving high-flow oxygen. It has been discussed elsewhere that the absence of a significant effect in this group was the result of remdesivir being prescribed too late after the onset of symptoms, but in the absence of validated published results, we chose to not take into account this potential effect.

When determining the parameters for the model, we systematically chose the most conservative options to not overrate the potential benefit of remdesivir and dexamethasone. As an example, there is currently an expert consensus indicating the use of more high-flow oxygen and less mechanical ventilation for critical COVID-19 patients, which in our model would increase the proportion of patients receiving remdesivir and therefore maximize its effect. However, there is still considerable variability in the scientific literature regarding the proportion of COVID-19 patients requiring mechanical ventilation, ranging from 29 to 94% [32]. Therefore, using the results provided by the ACTT-1 [5] limits the risk of overestimating the benefit of remdesivir, even potentially underestimating it. Similarly, we chose to set the number of hospitalized patients without O2 to 21% (hence, 79% in low-flow) even though most experts suggest that it is not more than 5%, as this would favour the impact of remdesivir, despite the lack of sufficient published data to support such a choice. In addition, we established an absence of the effect of remdesivir for patients with stays expected to be over 28 days. The length of stay duration reported in the ACTT-1 data was censored at 28 days, with some patients not discharged after 28 days. We conservatively assumed that no differences in the length of stay would be observed for patients discharged after 28 days. Based on the observed duration and hospital pathways in France, we assumed that these patients were likely to correspond to patients who will be secondary transferred to rehabilitation care when they survive. Moreover, no benefits on the length of stay in the ICU were assumed for patients for whom treatment with remdesivir had been initiated just before being transferred to the ICU following a worsening of their condition, thus potentially underestimating the benefits of remdesivir for these patients. Consequently, having made conservative choices in the hypothesis for our model, we are confident that we did not overestimate the impact of remdesivir but rather provided a lower bound estimate of its effect.

Medical innovation is by nature costly, and new treatments frequently come at a higher price, which should be balanced with the benefits obtained. The price of remdesivir may appear elevated until put in perspective with the cost of prolonged hospitalization for COVID-19 and the absence of alternatives with a proven antiviral effect. We did not provide an extensive cost-effectiveness analysis of remdesivir use, as we did not take into account associated

costs and loss of quality of life resulting from COVID-19. However, our budget impact analysis, by simply relating the cost of the treatment with the benefits gained by reducing the length of stay, showed that remdesivir use would result in savings that not only cover the price of the treatment but also generate gains. In the only scenario where there was no gain -the most conservative scenario-, the use of remdesivir was cost neutral.

Our study presents some limitations. The benefit of remdesivir and dexamethasone appears highly dependent on the timing of introduction following symptoms onset. However, in the absence of data, we were not able to assess the impact of this timing on the results. Therefore, it seems reasonable to highly recommend the early use of remdesivir and dexamethasone when indicated to maximize their effects. This would be before the 10th day after the onset of symptoms for remdesivir and one week after for dexamethasone. In addition, regarding the limited numbers of data sources available, we had to rely on only two RCT studies. It should be noted that we used the median time to recovery as a proxy for the length of stay in the absence of the availability of precise data. For the same reason, we did not use data from the SOLIDARITY trial [33] because data on the length of stay were not explicitly and precisely provided. Eventually, the standard of care dramatically changed between the ACTT trial period (March-April 2021) and the current period, with more patients being treated with high-flow oxygen or non-invasive ventilation instead of invasive ventilation. Our analysis supposes that the effect of remdesivir will remain unchanged on patients treated with high-flow oxygen, as we set the model's parameters according to the ACTT-1 patient distribution. Finally, recent German guidelines recommend the use of remdesivir for patients with low-flow oxygen therapy [34]. This is in accordance with our results, that remdesivir could be a tool to relieve hospitals by reducing bed-occupancy.

In conclusion, our results support the wide use of remdesivir among patients with low-flow and high-flow oxygen to flatten the epidemic curves and limit the saturation of hospital capacities. Even limited uptake or delayed introduction would result in a noticeable impact that would be beneficial for the patient, the hospital and the payor.

## Supporting information

**S1 Appendix.**
(DOCX)

## Acknowledgments

The authors thank American Journal Experts for reading and reviewing the original English-language text.

## Author Contributions

**Conceptualization:** Guillaume Béraud, Jean-François Timsit, Henri Leleu.

**Formal analysis:** Henri Leleu.

**Methodology:** Guillaume Béraud, Jean-François Timsit, Henri Leleu.

**Supervision:** Guillaume Béraud, Jean-François Timsit, Henri Leleu.

**Validation:** Guillaume Béraud, Jean-François Timsit, Henri Leleu.

**Writing – original draft:** Guillaume Béraud, Jean-François Timsit, Henri Leleu.

**Writing – review & editing:** Guillaume Béraud, Jean-François Timsit, Henri Leleu.

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
