## [Decision Letter · Decision Letter 0]

15 Jul 2021

PONE-D-21-17081

Remdesivir as a tool to relieve hospital care systems stressed by COVID-19: A modelling study on bed resources and budget impact.

PLOS ONE

Dear Dr. Beraud,

Thank you for submitting your manuscript to PLOS ONE. After careful consideration, we feel that it has merit but does not fully meet PLOS ONE’s publication criteria as it currently stands. Therefore, we invite you to submit a revised version of the manuscript that addresses the points raised during the review process.

ACADEMIC EDITOR:

This is a very intersting paper. Statistical analysis is well performed. Manuscript is presented in a standard English. There are minor conflicts between the reviews. Please answer to all the major and minor concerns moved. 

We look forward to receiving your revised manuscript.

Kind regards,

Martina Crivellari

Academic Editor

PLOS ONE

Journal Requirements:

"I have read the journal's policy and the authors of this manuscript have the following

competing interests:

- GB participated to advisory boards and gave lectures in symposia for Gilead.

- JFT reports participation to advisory board for Gilead related to covid-19 antiviral

therapies.

- HL is employed by PHE which received funding from Gilead."

Reviewers' comments:

Reviewer's Responses to Questions

**Comments to the Author**

1. Is the manuscript technically sound, and do the data support the conclusions?

Reviewer #1: Yes

Reviewer #2: Yes

Reviewer #3: Yes

Reviewer #4: Yes

2. Has the statistical analysis been performed appropriately and rigorously? 

Reviewer #1: Yes

Reviewer #2: Yes

Reviewer #3: Yes

Reviewer #4: Yes

3. Have the authors made all data underlying the findings in their manuscript fully available?

Reviewer #1: Yes

Reviewer #2: Yes

Reviewer #3: Yes

Reviewer #4: Yes

4. Is the manuscript presented in an intelligible fashion and written in standard English?

Reviewer #1: Yes

Reviewer #2: Yes

Reviewer #3: Yes

Reviewer #4: Yes

5. Review Comments to the Author

Reviewer #1: I found the approach of your article very interesting. As it seems, the use of Redemsivir on the right timing decreases hospital-bed occupancy in conventional wards and UCI: In this regard, we had the same perception in our department when we applied it during the second wave of the pandemic, as it allowed a faster turnover of patients.

To continue, I would like to give you some suggestions:

Introduction line 56: we know that clinical trials have inclusion criteria that generally limit validity to situations of routine clinical practice. The results in real life Will probably be more modest.

Line 61 It would be interesting to know what was the community transmissibility in those dates In order to to evaluate the output research in different areas with similar or different epidemiological indicators. I believe you should explain how was your department, area, hospital, Using some of these indicators:

• IA 14 days / 100000 inhabitants

• index R0.

• Total new cases per 100,000 persons in the past 7 days

• Percentage of NAATs that are positive during the past 7 days

Line 81: Although many of these variables that you mention work as indirect measures of transmissibility i would appreciate if you showed the cumulative incidence at 14 days during that pandemic period and the positivity rate of the tests (indicators that the ECDC, CDC uses to compare the risk of the different regions).

Line 194 The bootstrap is a statistical procedure used to approximate characteristics of the distribution in the sampling of a statistic. In this regard, a commonly used tool is simulation, as it generates a large number of samples through some type of resampling of the original sample. Its main advantage is that it does not require hypotheses about the generating mechanism of the data. Because of this I think the reader Would appreciate a brief description of what this “issue” consists of.

Line 204 Once again i insist that It would be helpful to know indicators of transmissibility in these areas at that time to compare effects with other regions and times.

Line 284 In the study, you descriptively evaluate an approximation to the cost in two scenarios. However, I think that studies designed with these objectives are necessary to demonstrate efficiency. It would be interesting to ask Whether the results of these models are confirmed taking into account all the costs.

Line 305 There are some papers regarding about effects of covid-19 in cancer delay, Myocardial infarct and stroke patients who can benefit from less ICU occupancy

As a last remark I think it is an article that provides a realization of a perception we had during the second wave Regarding the shortening of patients’ hospitalization.

Congratulations.

Reviewer #2: Advantages: 1. The manuscript has clear ideas and rigorous data. The predictive model considers the influencing factors more comprehensively, and the results obtained are representative.

2. The chart combines the very intuitive expression of the research purpose and the corresponding results.

Disadvantages: There are many researches on Radixivir, with mixed reviews. Although this manuscript has followed up with the research hotspots, the significance of the research results has not been well demonstrated in the article. It should be explained further.

Reviewer #3: I recommend this study for publication. However, I would ask you to consider the following points:

1) Please include the introduction epidemiological data from the context of the COVID-19 in France.

2) Please include a table with the probabilities of hospital admission, admission to the ICU and mortality adjusted for sex and comorbidities indicated in the methodology since it is important for the development of the model carried out.

3) In the subgroup of patients receiving high-flow oxygen or non-invasive ventilation (subgroup 6 of study ACTT-1) no statistically significant differences were observed in the length of hospital stay (Rate ratio: 0.82 (0.40–1.69) since 1 is within the confidence interval. Please, I propose to the authors to be conservative and not estimate this reduction in modeling.

4) Spend the next sentence of the methodology to the discussion or delete it: “This proportion was similar to the 35.2% that was seen in 150 a prospective cohort study that included 4,643 patients from 138 hospitals in France, Belgium 151 and Switzerland between February and May 2020 (21).”

5) Please discuss the results with other simulations carried out in other countries or other studies.

Reviewer #4: This is an interesting epidemiological modelling study where the purpose was to analyze the effects of remdesivir and dexamethasone, alone or in combination, on COVID-19 patient to reduce length of stay. They analyze the impact of a shorter time to recovery and a reduction in the risk of requiring high-flow or invasive ventilation associated with the use of remdesivir and dexamethasone. They also analyze the minimum uptake of remdesivir required to observe a significant impact on bed occupancy. They have used a calibrated and validated epidemiological model that was previously published.

Finally, they conclude that remdesivir use to high-flow-oxygen patients could have amplified reductions in ICU-bed and recommend the early use of remdesivir and dexamethasone when indicated to maximize their effects.

The study design is adequate to address the scientific question. The figures are adequate.

The authors studied the effectos of of remdesivir and dexamethasone, alone or in combination. However, in the title they include only remdesivir. Please, change the tittle by “Remdesivir and dexamethasone as a tool to relieve hospital care systems stressed by COVID-19:…”

6. PLOS authors have the option to publish the peer review history of their article (what does this mean?). If published, this will include your full peer review and any attached files.

Reviewer #1: No

Reviewer #2: No

Reviewer #3: No

Reviewer #4: No

---

## [Author Response · Author response to Decision Letter 0]

9 Aug 2021

Reviewer #1: I found the approach of your article very interesting. As it seems, the use of Redemsivir on the right timing decreases hospital-bed occupancy in conventional wards and UCI: In this regard, we had the same perception in our department when we applied it during the second wave of the pandemic, as it allowed a faster turnover of patients.

To continue, I would like to give you some suggestions:

We are grateful to the reviewer for the kind comments and the suggestions.

Introduction line 56: we know that clinical trials have inclusion criteria that generally limit validity to situations of routine clinical practice. The results in real life Will probably be more modest.

We agree with the reviewer that inclusion criteria in clinical trials provides an incomplete assessment of real-life results, which could result in more modest effects in real-life. However, modelling studies provide scenarios of what could be the results in real-life according to the chosen hypothesis. And by systematically choosing the most conservative options for the parameters, we thoroughly aimed at avoiding overrating the benefit of remdesivir. In that sense, we tried rather to provide a lower bound estimate of its effect and therefore are confident that results in real-life may also be more important.

Line 61 It would be interesting to know what was the community transmissibility in those dates In order to evaluate the output research in different areas with similar or different epidemiological indicators. I believe you should explain how was your department, area, hospital, Using some of these indicators:

• IA 14 days / 100000 inhabitants

• index R0.

• Total new cases per 100,000 persons in the past 7 days

• Percentage of NAATs that are positive during the past 7 days

We agree that providing some data on the pandemic in France at the time of the second wave is insightful, which is the reason why we provided the supplementary figure 1 and 2 for comparison as well as validation of the model. However, we provided graphs as summary of the data are less insightful and ours results are country-based and not specifically representative of a department, an area or hospital. With that in mind, we provided supplementary figures using the aforementioned indicators, in the appendix. Moreover, we’d like to highlight that the basic reproduction number R0 is not supposed to vary along the epidemic contrary to the effective reproduction number Re which changes all along the epidemic and is an indicator whether the epidemic is under control or not. But as Re could be estimated with various methods, and is not intrinsically included in an individual based model such as our, we chose to not provide it, as it wouldn’t be directly related to our study.

Line 81: Although many of these variables that you mention work as indirect measures of transmissibility i would appreciate if you showed the cumulative incidence at 14 days during that pandemic period and the positivity rate of the tests (indicators that the ECDC, CDC uses to compare the risk of the different regions).

Daily incidence is provided in supplementary figure 1, for model validation purpose. However, we acknowledge that a 14 days cumulative incidence with positivity rates would be informative, therefore we added such a figure in the appendix (supplementary figure 3).

Line 194 The bootstrap is a statistical procedure used to approximate characteristics of the distribution in the sampling of a statistic. In this regard, a commonly used tool is simulation, as it generates a large number of samples through some type of resampling of the original sample. Its main advantage is that it does not require hypotheses about the generating mechanism of the data. Because of this I think the reader Would appreciate a brief description of what this “issue” consists of.

Although there are different ways to proceed (parametric vs. non-parametric), a bootstrap procedure indeed consists in resampling the original dataset. We are not entirely sure of what the reviewer means by “description of what this issue consists of”. If the reviewer is expecting us to provide a brief explanation on the bootstrap procedure, then we added a couple of sentences in the discussion to expose what is bootstrap and its purpose regarding our study.

Line 204 Once again i insist that It would be helpful to know indicators of transmissibility in these areas at that time to compare effects with other regions and times.

As requested, we provided supplementary materials in the appendix for comparison purpose.

Line 284 In the study, you descriptively evaluate an approximation to the cost in two scenarios. However, I think that studies designed with these objectives are necessary to demonstrate efficiency. It would be interesting to ask Whether the results of these models are confirmed taking into account all the costs.

Cost-efficiency analysis is based on the comparison of 2 strategies (one of it could be doing nothing). Therefore, we indeed provided a cost-efficiency analysis, though a relatively simple one, to show that the benefit of shortening the length of stay would offset the cost of remdesivir. We are currently working on a more extensive cost-efficiency analysis, taking into account indirect and secondary costs etc. However, precise and valid data on indirect costs are still scarce at the moment, and including all the possible related costs would probably be beneficial to remdesivir. As we aimed at being conservative, we chose to avoid providing a cost-efficiency analysis with too many parameters and too much uncertainty, as it could overestimate the benefit of using remdesivir. But that would be the next step as soon as the current study is published.

Line 305 There are some papers regarding about effects of covid-19 in cancer delay, Myocardial infarct and stroke patients who can benefit from less ICU occupancy

We fully agree with the reviewer, which is why we cited one study on cancer diagnosis delay, and another on cardiac arrest, but also knowing that there will be more conclusive studies in the near future providing precise data that could be implemented into the model.

As a last remark I think it is an article that provides a realization of a perception we had during the second wave Regarding the shortening of patients’ hospitalization.

Congratulations.

Reviewer #2: Advantages: 1. The manuscript has clear ideas and rigorous data. The predictive model considers the influencing factors more comprehensively, and the results obtained are representative.

2. The chart combines the very intuitive expression of the research purpose and the corresponding results.

Disadvantages: There are many researches on Radixivir, with mixed reviews. Although this manuscript has followed up with the research hotspots, the significance of the research results has not been well demonstrated in the article. It should be explained further.

We are grateful to the reviewer for the comment. The purpose of this study is clearly stated from the title, an assessment of the impact of remdesivir use on bed-occupancy. For that, we based our study on the reduction of hospital length, which is a point that has not been invalidated among the numerous studies already published to date. Indeed, recent German guidelines recommend the use of remdesivir with the clear purpose of relieving hospital system by shortening length of stay. As requested, we developed this aspect in the manuscript at the end of the discussion.

Reviewer #3: I recommend this study for publication. However, I would ask you to consider the following points:

1) Please include the introduction epidemiological data from the context of the COVID-19 in France.

As requested, and in accordance with the requests from reviewer #1, we added some graphs in the appendix, to facilitate comparison and to expose the epidemiological situation in France.

2) Please include a table with the probabilities of hospital admission, admission to the ICU and mortality adjusted for sex and comorbidities indicated in the methodology since it is important for the development of the model carried out.

These data are already included in the appendix, with the appropriate references. Impact of sex and comorbidities were taken into account for outcomes as mentioned, but not for hospitalization. With these data and the model already published, anyone should be able to reproduce our results and to develop it further.

3) In the subgroup of patients receiving high-flow oxygen or non-invasive ventilation (subgroup 6 of study ACTT-1) no statistically significant differences were observed in the length of hospital stay (Rate ratio: 0.82 (0.40–1.69) since 1 is within the confidence interval. Please, I propose to the authors to be conservative and not estimate this reduction in modeling.

We agree that patient receiving high-flow or non-invasive ventilation benefits differently from remdesivir compared to low-flow patients, which is one of the reasons why we distinguished the 2 groups. However, the rate ratio mentioned (0.82 (0.40–1.69)) is for mortality through day 14, not the length of stay. We based our calculation on the reduction on the median time to recovery from 20 to 15. It should also be noted that we didn’t base our calculation on the number of recoveries but on the reduction in the length of stay. Unfortunately, more precise data were not available. We developed this aspect in the following paragraph: It is important to stress that the decrease in bed occupancy came from a shorter time to recovery but not from a higher rate of recovery, which was not significantly different between remdesivir and placebo for patients receiving high-flow oxygen. It has been discussed elsewhere that the absence of a significant effect in this group was the result of remdesivir being prescribed too late after the onset of symptoms, but in the absence of validated published results, we chose to not take into account this potential effect.

4) Spend the next sentence of the methodology to the discussion or delete it: “This proportion was similar to the 35.2% that was seen in 150 a prospective cohort study that included 4,643 patients from 138 hospitals in France, Belgium 151 and Switzerland between February and May 2020 (21).”

We spent the sentence to the discussion as requested, as we though important to provide an external validation to our hypothesis.

5) Please discuss the results with other simulations carried out in other countries or other studies.

To the best of our knowledge, there is no published modelling studies on the impact of remdesivir on bed-occupancy so far, therefore we were not able to compare our results. But we are nonetheless confident that our article will favour many others studies carried out in different contexts.

Reviewer #4: This is an interesting epidemiological modelling study where the purpose was to analyze the effects of remdesivir and dexamethasone, alone or in combination, on COVID-19 patient to reduce length of stay. They analyze the impact of a shorter time to recovery and a reduction in the risk of requiring high-flow or invasive ventilation associated with the use of remdesivir and dexamethasone. They also analyze the minimum uptake of remdesivir required to observe a significant impact on bed occupancy. They have used a calibrated and validated epidemiological model that was previously published.

Finally, they conclude that remdesivir use to high-flow-oxygen patients could have amplified reductions in ICU-bed and recommend the early use of remdesivir and dexamethasone when indicated to maximize their effects.

The study design is adequate to address the scientific question. The figures are adequate.

The authors studied the effectos of of remdesivir and dexamethasone, alone or in combination. However, in the title they include only remdesivir. Please, change the tittle by “Remdesivir and dexamethasone as a tool to relieve hospital care systems stressed by COVID-19:…”

We modified the title as requested.

---

## [Decision Letter · Decision Letter 1]

24 Dec 2021

Remdesivir and dexamethasone as tools to relieve hospital care systems stressed by COVID-19: A modelling study on bed resources and budget impact.

PONE-D-21-17081R1

Dear Dr. Béraud,

We’re pleased to inform you that your manuscript has been judged scientifically suitable for publication and will be formally accepted for publication once it meets all outstanding technical requirements.

Kind regards,

James Mockridge

Staff Editor

PLOS ONE

Reviewers' comments:

Reviewer's Responses to Questions

**Comments to the Author**

1. If the authors have adequately addressed your comments raised in a previous round of review and you feel that this manuscript is now acceptable for publication, you may indicate that here to bypass the “Comments to the Author” section, enter your conflict of interest statement in the “Confidential to Editor” section, and submit your "Accept" recommendation.

Reviewer #1: All comments have been addressed

Reviewer #2: All comments have been addressed

Reviewer #3: All comments have been addressed

2. Is the manuscript technically sound, and do the data support the conclusions?

Reviewer #1: Yes

Reviewer #2: Yes

Reviewer #3: Yes

3. Has the statistical analysis been performed appropriately and rigorously? 

Reviewer #1: Yes

Reviewer #2: Yes

Reviewer #3: Yes

4. Have the authors made all data underlying the findings in their manuscript fully available?

Reviewer #1: Yes

Reviewer #2: Yes

Reviewer #3: Yes

5. Is the manuscript presented in an intelligible fashion and written in standard English?

Reviewer #1: Yes

Reviewer #2: Yes

Reviewer #3: Yes

6. Review Comments to the Author

Reviewer #1: I believe that the author has adequately collected the recommendations and complies with ethical aspects about research and publication.

Reviewer #2: First of all, thank you for allowing me to review this manuscript again. On the whole, this revision was completed on time according to my expectations and suggestions.

Reviewer #3: Thank you for reviewing and including the comments, I think it is a great work that provides evidence on the current uncertainty of treatments for COVID-19.

7. PLOS authors have the option to publish the peer review history of their article (what does this mean?). If published, this will include your full peer review and any attached files.

Reviewer #1: No

Reviewer #2: No

Reviewer #3: No

---

## [Editor Report · Acceptance letter]

3 Jan 2022

PONE-D-21-17081R1 

Remdesivir and dexamethasone as tools to relieve hospital care systems stressed by COVID-19: A modelling study on bed resources and budget impact. 

Dear Dr. Béraud:

I'm pleased to inform you that your manuscript has been deemed suitable for publication in PLOS ONE. Congratulations! Your manuscript is now with our production department. 

Kind regards, 

on behalf of

Dr James Mockridge 

Staff Editor

PLOS ONE